

# Estimating water residence time distribution in river networks by boosted regression trees (BRT) model

Meili Feng [1, 2, 3], Martin Pusch[2], Markus Venohr[2]

[1]School of Geographical Sciences, University of Nottingham Ningbo China, Ningbo, 31500, China
[2]Leibniz-Institute of Freshwater Ecology and Inland Fisheries, Berlin, 12587, Germany
[3]Department of Civil, Environmental and Mechanical Engineering, University of Trento, Trento, 38123, Italy

*Correspondence to*: Meili Feng (meili.feng@nottingham.edu.cn)

**Abstract.** In-stream water residence time (WRT) in river networks is a crucial driver for biogeochemical processes in riverine ecosystems. Dynamics of the WRT are critical for understanding and modelling nutrient retention in lakes and rivers, in particular during flood events when riparian areas are inundated. This study illustrates the potential utility of integrating spatial landscape analysis with machine learning statistics to understand the effects of hydrology and geomorphology on WRT in river networks, especially at large scales. We applied the Boosted Regression Trees (BRT) approach to estimate water residence, a promising multi-regression spatial distribution model with consistent cross-validation procedure, and identified the crucial factors of influence. Reach-average WRTs were estimated for the annual mean hydrologic conditions as well as the flood and drought month, respectively. Results showed that the three most contributing factors in shaping the WRT distribution are river discharge (57%), longitudinal slope (21%), and the drainage area (15%). This study enables the identification of key controlling factors of the reach-average WRT and estimation of WRT under varying hydrological conditions. The resulting distribution model of WRT is an easy to apply and sound approach helping to improve water quality modelling at larger scales and water management approaches aiming to estimate nutrient fluxes in river systems.

**Keywords:** Water residence time; river networks; spatial distribution model; Boosted Regression Trees (BRT).

## 1 Introduction

Water residence time (WRT) (also known as in-stream water residence time, Worral et al., 2014) refers to the average time that a certain amount of water travels through a defined river reach. Reach-average WRT represents one of the most important determinants for in-stream biogeochemistry recycling processes (Catalán et al., 2016; Drummond et al., 2016; Ensign and Doyle, 2006; Gibson, 2000; Hrachowitz et al., 2016; Stanley and Doyle, 2002). The response of river flow to precipitation is highly nonlinear, and so are the in-stream processes of water retention (Heidbüchel et al., 2012). Water residence time in river networks differs due to the variability of inflow rates, river topologic and geomorphologic parameters such as slope (Doyle et al., 2005; Wang et al., 2015). Residence time studies especially for extreme hydrologic conditions (i.e. flood and drought events) are of particular importance for water management practice as here significant share of annual fluxes can be transported or retained during a short period of time.



Studies on WRT are often based on process-based, deterministic models for hydrological processes including groundwater, precipitation and surface runoff in a river basin (i.e. SWAT (Grizzetti et al., 2003), SPARROW (Preston Seitzinger et al., 2011)). However, these deterministic models are data demanding and time consuming when applying to networks of large

river systems. Besides modelling, another widely applied way of measuring water residence time is by introducing solutes and measured residence time and flow velocity within specific river reach. WRT can be estimated based on the travel time of dissolved solute tracers that are experimentally added to the river, which have been used to analyze retention efficiency and up-take length of dissolved nutrients may be retained by bio-chemical processes (Drummond et al., 2016; Nieuwenhuyse, 2005; Soulsby et al., 2006). Further improvements of the process-based models will require addressing spatial heterogeneities

within basins (Mayorga et al., 2010) and a better understanding of river network retention and the driving factors controlled by runoff within watershed (Arora et al., 2016; Dumont et al., 2005).

Computational and empirical methods (e.g. MONERIS, Venohr et al., 2011) offer diversified options in combining them to statistical and process-based models at different scales (Gottschalk et al., 2006; Nieuwenhuyse, 2005; Soulsby et al., 2006). The one-dimensional hydraulic Manning-Strickler formula, which calculates flow velocity in dependence of channel slope and

cross-section variations, has been widely used to estimate flow velocity and thus water residence time (Verzano et al., 2012; Worrall et al., 2014). However, distributed data to model the WRT in river networks, especially at larger scales, is often lacking or only available in insufficient spatial resolution. Consequently, any modelling approach addressing WRT or related in-stream processed at larger scales is limited by the quality of observed or alternatively on estimated data on geomorphological (roughness, slope, sediment) parameters. The estimation of hydrological regimes in complex river systems still remain

controversial between detailed process-based models on one side, and over simplified empirical methods on the other side. This gap appears to be even larger when it comes to the application of large-scale river basins.

To improve the understanding of WRT as carrier and as driving force for instream processes, while considering impacts hydro-morphological of river channels characteristics (Poole, 2010), Behrendt and Opitz (1999) discussed the dependence of nutrient retention on specific runoff and hydraulic load in river systems. Gücker and Boëchat (2004) investigated the ammonium

retention in tropical headwater streams with differing channel morphology and hydraulic characteristics such as riffles and pools. Doyle and Stanley (2003) highlighted the controlling position of hydro-geomorphological factors in nutrient cycling processes. Helton et al. (2018) reemphasized the importance of structures of the whole stream network in nitrogen transformation and removal with varied but integrated spatial distribution from headwaters to downstream. To determine how differences in geomorphologic settings influence spatial heterogeneity in transport and retention of nutrients, a hierarchical

network perspective is needed, comprising connectivity, residence times, and reactivity interactions (Lin et al., 2016; Stewart et al., 2011).

Beyond the traditional insights of nonlinear processes using 1-D, 2-D or 3-D hydrodynamic equations, other nonlinear statistical approach such as the Boosted Regression Trees (BRT) is becoming to play a part in hydrodynamic studies (Ouedraogo and Vanclooster, 2016; Toprak et al., 2014; Toprak and Cigizoglu, 2008). The BRT model, which combines

advantages of regression trees and boosted adaptive method, has recently been applied in studies on ecological traits and



species distributions (Elith and Leathwick, 2017; Wyse and Dickie, 2017; Zimmermann et al., 2010), as well as in other environmental research fields, such as natural flow regimes, groundwater and hydraulic conductivity (Jorda et al., 2015; Mousavi et al., 2017; Naghibi et al., 2016; Snelder et al., 2009), soil science (Martin et al., 2009; Jalabert et al., 2009), air pollution (Carslaw et al., 2009), energy (Kusiak et al., 2010), or climate change (Shabani et al., 2016) and so on. With consistent

cross-validation procedure and the feature of easy application, the BRT model provides a strong potential for applying large-scale WRT analysis while considering multiple hydro-geomorphological parameters.

In this study, we investigate the spatial distribution of water residence time across a wide range of hydro-geomorphologic settings by applying the machine-learning approach known as boosted regression trees (BRT). The selection of leading indicators for predicting the reach-average WRT in 82 river networks in Germany are analyzed.

## 2 Methods and Materials

### 2.1 Study area and dataset

We collected discharge data for the years 2008-2014 from 132 gauging stations in Germany that are recorded with a temporal resolution of 15 minutes (Bavarian State Office for the Environment, and the ITZ Bund www.pegelonline.wsv.de). Using these stations, 82 river reaches were identified (Figure 1), which are delimited at both (upstream and downstream) ends by gauging

stations. These reaches are geographically widely distributed across Germany, although underrepresenting northern low lands, and represent the hydro-morphological conditions of 13 stream types that differ in their biogeochemical conditions, too. To be noted, discontinuities in the river systems, as lakes and impoundments (produced by weirs or dams) are not explicitly considered in this paper, as water residence time is much longer and is controlled by other aditional mechanisms such as the features of stratification, water body internal currents and volume changes by input-output controls (Heidbüchel et al., 2012;

Ji, 2008; Rueda et al., 2006).

### 2.2 Sources for factors assumed to affect water residence time

In this paper we evaluate the average discharge, drainage area, mean river width, length, and slope, along with sediment composition as hydro-geomorphologic factors for predicting WRT in the selected river reaches. Geomorphologic parameters are averaged over the reach between the upstream and downstream stations to represent the mean situation of the selected river

reach (Table 1). Substrate class of the sediment type for each river reach is represented in percentage (100% all classes in sum) according to their length that falls into each class. According to the German soil classification system (Working Group on Soil Classification of the German Soil Science Society, 1997), six substrate classes are derived as Sand (S), Clay (C), Silt (U), Loam (L), Peat bog (HM), Fen (NM), respectively. Furthermore, classifications of stream types with integrated geomorphic features (Table 2) are applied as an indicator for specific landscape characteristics. The stream types are aligned with the

official German stream and river type classification systems based on physio-chemical parameters and geological classifications of ecological zones that contain similar environmental characteristics, such as stream size, stream order, altitude,





flow velocity, streambed substrates, water temperature, or stream width etc. (Pottgiesser and Sommerhäuser, 2004; Arle et al., 2014). All geographic analyses and calculations were performed in ArcGIS Desktop (Version 10.0, Environmental Systems Research Institute (ESRI). Redlands, CA, USA).

## 2.3 Spatial distributions of water residence time

We introduced the nonmetric multidimensional scaling (NMDS) plots (Agarwal et al., 2007) to obtain insight into the patterns of hydro-morphological conditions as well as WRT distributions for the studied river reaches. NMDS allows integrating different data formats, such as continuous monitoring data, discrete parameters, binary data or binomial category datasets. We used the Gower's generalized coefficient of dissimilarity approach (Gower and Legendre, 1986) to standardize the continuous variables against the discrete ones and got the standardized Euclidean distance for the NMDS plots.

Water residence time for studied river reaches are estimated for the average discharge conditions during 2008-2013. The spatial distribution of water residence time has been estimated by applying the nonlinear model of Boosted Regression Trees (BRT). The feature of nonlinearity in controlling WRT, and interactions among multiple predictive factors are analyzed via the BRT model (Elith and Leathwick, 2016). In the first place, the fitted model is generated by the known average values of predictive datasets for the complete river networks. Furthermore, the generated model is applied to estimate the distribution of WRT under different environmental scenarios such as flood or drought hydrologic conditions. A measurement of relative importance (in percentage) is calculated by the model to facilitate the comparisons of term-wise contributions. In addition, partial dependence plots and fitted link functions for each variable were produced. Fitted BRT models were obtained by the sum of all trees multiplied by the learning rate (Elith et al., 2008):

$$f(x)=g[\sum_i T_i(x)] \tag{1}$$

where f is the fitted model, x is the independent variable, $T_i$ are the individual learners, and $g$ is the link function that grows optimum trees. All calculations and modelling were performed in R (R Core Team, 2016) by using the package 'dismo' (Hijmans et al., 2016) and 'gbm' (Greg Ridgeway with contributions from others, 2015).

## 2.4 Model validation

We here applied the empirical equation of previous studies $t = aQ-bxc$ proposed by Graf (1986), in which $t$ represents the water residence time, $Q$ is the discharge, $x$ is the traveled distance in downstream direction, and $a, b, c$ are the coefficients.

To compare the results of WRT estimated by the BRT model (WRT$_{pred}$) against those of the empirical equations and observed values, the error of the prediction is calculated using the Root Mean Squared Errors (RMSE):

$$RMSE = \sqrt{\frac{\sum_{i=1}^n (WRT_{pred}-WRT_{obs})^2}{n}} \tag{2}$$



where $WRT_{pred}$ is the predicted water residence time (h/km) and $WRT_{obs}$ is the original calculated value according to observation

at the river reach of i, and n is the number of studied river reaches.

## 3. Results

The elaboration of the results is structured in a) the spatial dissimilarity of geomorphology and hydrological factors for the

studied river reaches, followed by b) the results of relative importance of variables calculated by the BRT model. Furthermore,

we discussed the spatial distribution of estimated WRT under long-term annual average discharge conditions as well as during

the extreme hydrological month of flood and drought.

### 3.1 Governing factors for water residence time

The multidimensional Euclidean distance between the studied river reaches derived from varying hydro-geomorphology

channel characteristics is shown in the Nonmetric Multi-Dimensional Scaling (NMDS) plot (Figure 2). River reaches are

grouped and colored according to their stream type classification and revealed clustering patterns in agreement with the river

size. The river type classification of ecoregion independent streams include lake outlets (type 21) and small organic substrate-

dominated rivers (type 11).

Fitted BRT models were obtained by the sum of all trees multiplied by the learning rate of each predictive variables. The fitted

model accounted for 54.5% of the mean total deviance of the monitored dataset (1-mean residual deviance / mean total

deviance). The optimal fit was achieved with the following variable setting: interaction depth = 10, tree complexity = 10,

learning rate = 0.001, bag fraction = 0.5 and cross-validation = 10-folds, optimal number of trees = 1680. For this fit, the

training data correlation coefficient was 0.668, and cross-validation correlation coefficient was 0.614.

The predictive variable of mean discharge represented the most influential variable (57.4%) in the BRT model, followed by

slope (21.5%) and the sum of drainage area (15.6%). Mean river width and river types together only explained less than 4%

to the model variance. Similarly, substrate classes did not significantly influence water residence time (< 2%). In particular,

the substrates of clay, peat bog and fen showed zero statistical contribution (Table 3). Although the latter predictive variables

have little or no importance in our study, we did not exclude them from the set of the predictive variables while remaining the

complete dataset for further analyses of any other scenarios.

After accounting for the average effects of the predictive variables for all river reaches using the BRT model, we used partial

dependence plots to identify the relative influence of the dominant eight variables on $WRT_{pred}$ in the individual reaches (Figure

3). The y-axis shows the fitted $WRT_{pred}$, optimized on basis of link function Eq. (1). At low values, mean discharge, drainage

area and width (statistically not significant) has a strong negative impact on WRT. The influence of these three predictive

variable decrease with increasing respective values and only cause smaller changes in WRT. Longitudinal slope of riverbed is

found to have a positive relationship with WRT. One possible reason for this is, that with slope also turbulence increases, but





not the mean flow velocity. This is revealed by the less dominant position of slope in comparison with discharge, area and
river width. The effects of river topography and sediment composition appear to be largely mediated by their implemented
correlation with hydrological characteristics and geographic distribution.

### 3.2 WRT distribution

### 3.2.1 WRT distribution under annual average hydrologic conditions

River reaches categorize as large rivers (type 9.2, 10, 15, 20) usually have high levels of the average discharge and area as
well as low slope values, which consequently show a synchronized distributions in the NMDS plot (Figure 4). Water residence
time at smaller rivers is in turn stronger impacted by distinct topologic features, resulting in a less uniform pattern than the
group of large river.

Scattered from the Euclidian distance to the spatial dimension, water residence time distribution under the mean discharge
conditions during 2008-2013 are displayed Figure 5. River discharge above 300 m3/s results in WRT of less than 4 h/km,
which equals to flow velocity of more than 0.07 m/s. Comparing the results of BRT model (WRT$_{pred}$) with that of the observed
values (WRT$_{obs}$) showed 46% of the mean squared errors are less than 0.1 h/km (Figure 6). Poor model performance under
low flows demonstrates the increased impact of individual predictive variables and the need for further testing and data
collection to support the inclusion of additional biogeochemistry processes. Site-specific uncertainties might arise from
unknown flow paths and mixing dynamics that significantly affect management strategies.

Through comparing the results of predicted WRT (WRT$_{pred}$) with that of Graf's empirical equation (WRT$_{emp}$), the calculated
flow velocity by BRT model showed a less correlated linear relationship with discharge (Figure 7). WRT$_{pred}$ and WRT$_{emp}$
delivered water residence times in the same magnitude of order, but with different gradients in their variability and dependence
with changing Qmean. A lower tendency of a linear relationship for smaller discharge levels below 500 m3/s indicates a
potential geomorphological influence manifested at small rivers and non-bankfull conditions. Possible explanations for the
described differences between WRT$_{pred}$ and WRT$_{emp}$ could be that in this study, the BRT model is built to explain variables
through multiple boosted regressions by including the nonlinear interactional effects among predictive variables. This
interpretation is in accordance with the partial dependency analysis of each variable that an overall consideration of all
predictive variables at varied levels are needed by applying the systematic or network approach (Dumont et al., 2005).

### 3.2.2 WRT distribution under hydrologic extremes

The response of water residence time on changing and in particular extreme discharge is complex, especially for distinct
geomorphic sites. In order to facilitate more intuitive understanding, we did parallel studies for the extreme flood event in June
2013 and the driest month of November 2011 in Germany. The May/June 2013 flood was the most severe large-scale flood
events in Germany during the last 6 decades (Merz et al., 2014). Compared with the flood events in June 2013, the median
discharge in November 2011 is 80.2% lower with the estimated water residence time is 20.7% (0.17 hours) longer per kilometer



(Figure 8). Spatial variation is shown through the bivariate map of mean discharge and water residence times. The contrasting effect is more clearly observed in the Elbe river basin where the most severe floods occurred (Figure 9).

## 4. Discussions

### 4.1 Interactional effects of predictive variables

Getting to know the interactional effects among predictive variables would facilitate the empirical estimation of WRT in a river reach on basis of generally available information. Among all the predictive variables, river hydrology ranks the first place of relative importance together with slope in shaping the variation of water residence time. Therefore, hydrological variations in the river reach have to be the paramount element of discussion.

The 2-dimensional partial dependence plot in Figure 10 shows the interactional effects between river discharge and drainage
area. Another important geomorphological factor is the river width that has great contribution to the distribution of WRT. Furthermore, the river type classification, which implements generalized geomorphologic and geographic attributes, could act as a substitute, helping to simplify the process of WRT estimation especially under limited data availability. The interactional effect between river type and mean discharge is expressed in Figure 11. For a river reach with known substrate class and river topology, water residence time under different discharge levels can be estimated.

### 4.2 Impacts of groyne fields on water residence time

As mentioned earlier in methodology part, the discontinuities in the river systems, potentially impacting water residence time, such as lakes, impoundments or river groyne fields were not included in the original dataset for model generation. River groynes (also called wing dams) usually made from rocks or woods are constructed from the riverbank, transversely to the main flow direction to prevent from lateral bank erosion by reducing flow velocity (Yossef, 2002). Due to simple construction,
long-term durability and major functions, groyne fields (GF) are very widely applied in the lowland rivers of Germany. At present there are approximately 6900 groynes, covering 92% of the banks along the Middle Elbe River section (Schwartz, 2006). Because of the considerable reduction of water depth and flow velocity relative to the main stream, the prolonged retention time of water in the GF has important impact on the nutrient uptake dynamics and phytoplankton growth (Engelhardt et al., 2004; Guhr et al., 2000; Ockenfeld and Guhr, 2003). Hydraulic waves attenuation and increasing water residence time
are two main effects of groyne fields and potentially have strong impacts on nutrients retention and phytoplankton growth. Describing the specific hydraulic characteristics of flow velocity and residence time patterns in GF is the key for understanding the ecological significance of these retention zones.

At 14 out of 82 studied river reaches groyne fields are installed. Distribution of water residence time at groyne fields are linked to the variables and factors as discussed above, however, the shapes of the hydrograph reveal the different attributes of
attenuation, which indicate the potential for nutrient retention. In order to exclude the influence of distinct scales, river reaches



from the alpine stream and central highlands in Bavaria (in total 39 reaches) are not considered for the comparison. Among the rest 43 river reaches of comparable discharge level, two groups of 14 river reaches with groyne fields and 29 free-flowing rivers are compared.

We plot the cumulative distribution functions of the mean hydro-width (in hours) for the two groups (Figure 12). The cdf plot shows that the probability level of hydro-width less than 90% are up to 44.4 hours at groyne fields, compared with that of only 21.4 hours at free-flowing rivers. There is very little chance (< 2.5 %) that the probability of hydro-width in free-flowing rivers will be less than one hour and there is also small chance (< 5%) that it could be as high as 33.7 hours. The groyne fields show pronounced wider hydro-width than the free-flowing rivers: with 59.4% (87.6 hours) larger maximum value and more than 2 folds' (4.7 hours) at the median level. Not surprisingly, the estimated water residence time for groyne fields show higher probabilities for WRT less than 1.5 h/km in comparison with the free flow rivers (Figure 13).

**4.3 Hydraulic residence time as a lens for nutrient retention time**

It should be noted of the existing differences between flow velocities in the system (that set the velocity of conservative solutes) and the celerity (or speed with which hydraulic perturbations are conveyed, which control the hydrograph), are to be the velocity of conservative solutes, expected since they are controlled by different mechanisms. The nutrient transportation velocity in streams is always slower than the kinematic flow celerity of gravity-driven hydraulic waves. Studies on the differentiation and translation of these two velocities under varying flow conditions have been thoroughly discussed by McDonnell and Beven (2014). The water residence time discussed in this paper is coherently referred to flow velocity. However, the biogeochemical functioning of a river ecosystem is largely dependent on the transportation processes of water and dissolved substances within the geomorphic context of river networks (Benettin et al., 2015; Withers and Jarvie, 2008). Nutrient dynamics are controlled by the interaction of several key parameters, i.e. river discharge, channel geometry and vertical exchanges of water (Maazouzi et al., 2013). The amount of biologically labile dissolved nutrient effectively taken up by primary producers depends, next to other limiting factors, such as light availability or water temperature, on the WRT in a river section during which processes may take place (Drummond et al., 2016; Ensign and Doyle, 2006; Nieuwenhuyse, 2005; Soulsby et al., 2006).

Water flow velocity in a river reach controls the time during which bio-chemical processes can reduce or transform contained nutrients. Nutrient transport and transformation in streams involves both physical dynamics and biological uptake processes along the longitudinal course of rivers (Kronvang et al., 1999; Runkel, 2007). The transport mechanisms are mainly shaped by hydro-morphological parameters such as river discharge, water depth and velocity, and by other related physical ones such as sediment composition. River hydro-morphological shapes those processes and plays a major role in structuring the hydrological, ecological and biogeochemical dynamics in streams and rivers that are essential to ecosystem functioning (Doyle et al., 2003).



Traditional insights into the processes of nutrient spiraling process by experiments are sometimes biased and conditional dependent due to the fact that nutrient addition often brings much higher concentration than the background level, which results with overestimated nutrient uptake length (Mulholland et al., 2002). Modelling studies on nutrient export are mostly based on
steady state hydrologic conditions assuming variations in pressure from pollution sources (Ingestad and Ågren, 1988; Powers et al., 2009; Runkel, 2007; Runkel and Bencala, 1995). This assumes that the hydraulic gradients that drive the transports are maintained the whole time the stream water remains in the water body, which is unrealistic (McCallum and Shanafield, 2016). Therefore, the current study adds to improve our understanding on the impact of water residence time on nutrient retention time and transportation processes on medium to large scales. This is in particular required to tackle the challenge in quantifying
changes of hydro-morphological parameters in space and time (Ambrosetti et al., 2003; Bouwman et al., 2013; Tong and Chen, 2002).

## 5. Conclusions

Understanding the dynamics of instream water residence time does not only assist water quality and instream nutrient retention modelling, but also water management practices especially under extreme hydrologic conditions. Through application of the
BRT model for estimating WRT in river networks, we identified that river discharge weights the most compared with river topologic and geomorphic attributes. At the scale of river networks, water residence time is primarily affected by river discharge, followed by river width and river channel slope. Geomorphological attributes are more influential on small rivers in the Alpine mountainous areas. By taking the example of river discharge during flood and drought events, the BRT modelling is useful for water residence time estimation under extreme hydrological scenarios.

We conclude that the BRT approach has the potential to be used for addressing how timescales of the hydrological cycle change at different scales. The spatial distribution model contributes to an advanced methodology in WRT estimation in between of complex deterministic process models and empirical statistical models, and can be applied to study areas of diversified scales. The research element of water residence time is elaborated on the temporal scale of multi decades, though it is based on reach-scale information. It has the foremost implications on river ecology by linking nutrient retention time in
streams at larger spatial and temporal scales. While applied to few representative catchments. It can set the basis for analyses of large spatial scales and can be relevant at longer time scales associated with climate variability.

The results underline the relative importance of hydro-geomorphological features, which has clear implications to optimize the efficiency of river restoration effects on runoff processes. In combination with developed nonlinear spatial statistics it could be a novel approach in solving hydro-geophysical or even social economic distribution related questions. A further
investigation of retention time maxima could help identifying thresholds at which potential restoration measures, land use changes, drought or floods, and climatic stressors affect physical water body conditions and bio-chemical responses.





## Acknowledgements

This work has been carried out within the SMART Joint Doctorate programme "Science for the MAnagement of Rivers and their Tidal systems" funded by the Erasmus Mundus program of the European Union. The authors thank the Bavarian State Office for the Environment to provide hydrological dataset for Bavaria, and the ITZBund for providing hydrological monitoring dataset for Germany (www.pegelonline.wsv.de). A special thanks to Judith Mahnkopf and Annett Wetzig in the working group of "nutrient balances in river systems" at the Leibniz Institute of Freshwater Ecology and Inland Fisheries Berlin for their support with environmental data collection and preparation. The authors thank the colleagues and friends who offered their valuable comments and proof readings for the manuscript.

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





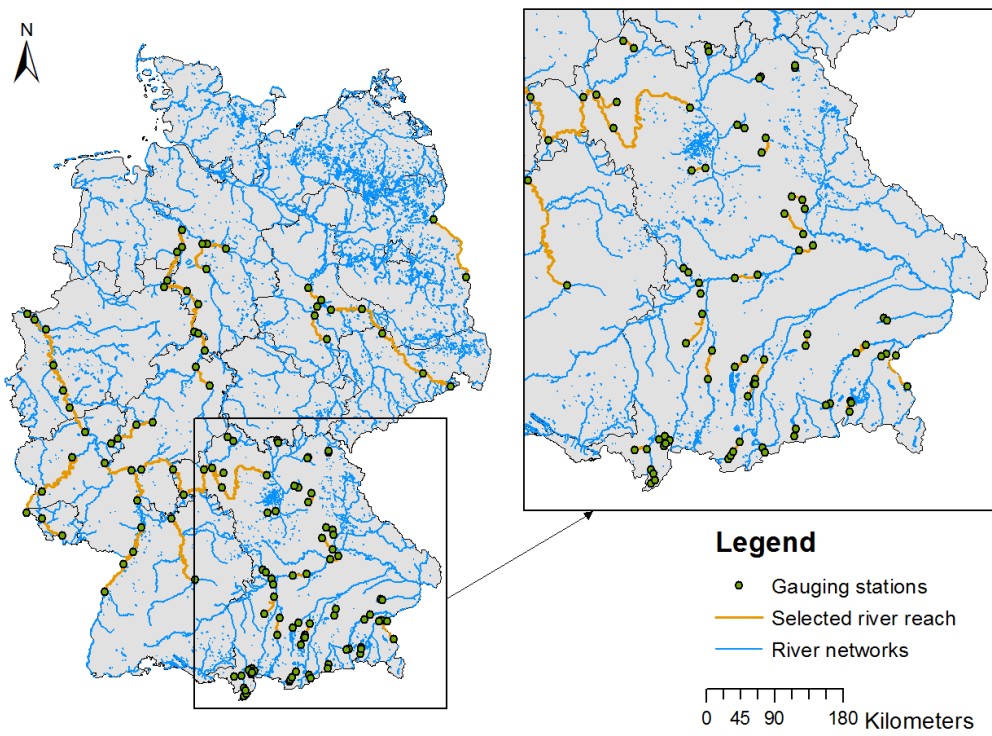


**Figure 1: Map of river networks in Germany, with selected river reaches (orange) and corresponding upstream-downstream gauging stations (circles).**






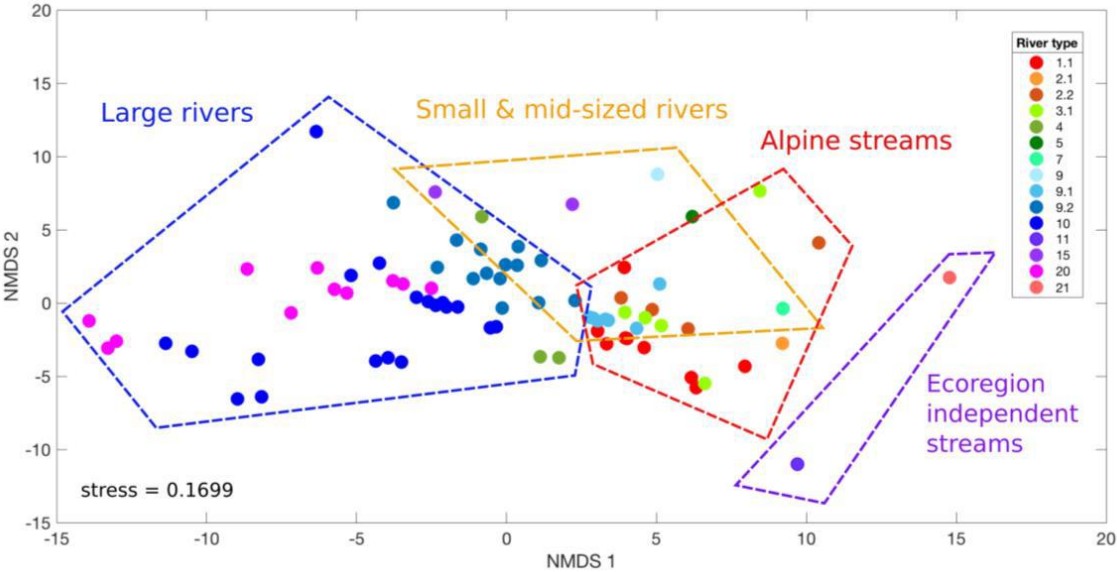

**Figure 2: Dissimilarities of the studied river reach in the Nonmetric Multi-Dimensional Scaling (NMDS) ordination space according to hydro-geomorphic attributes.**



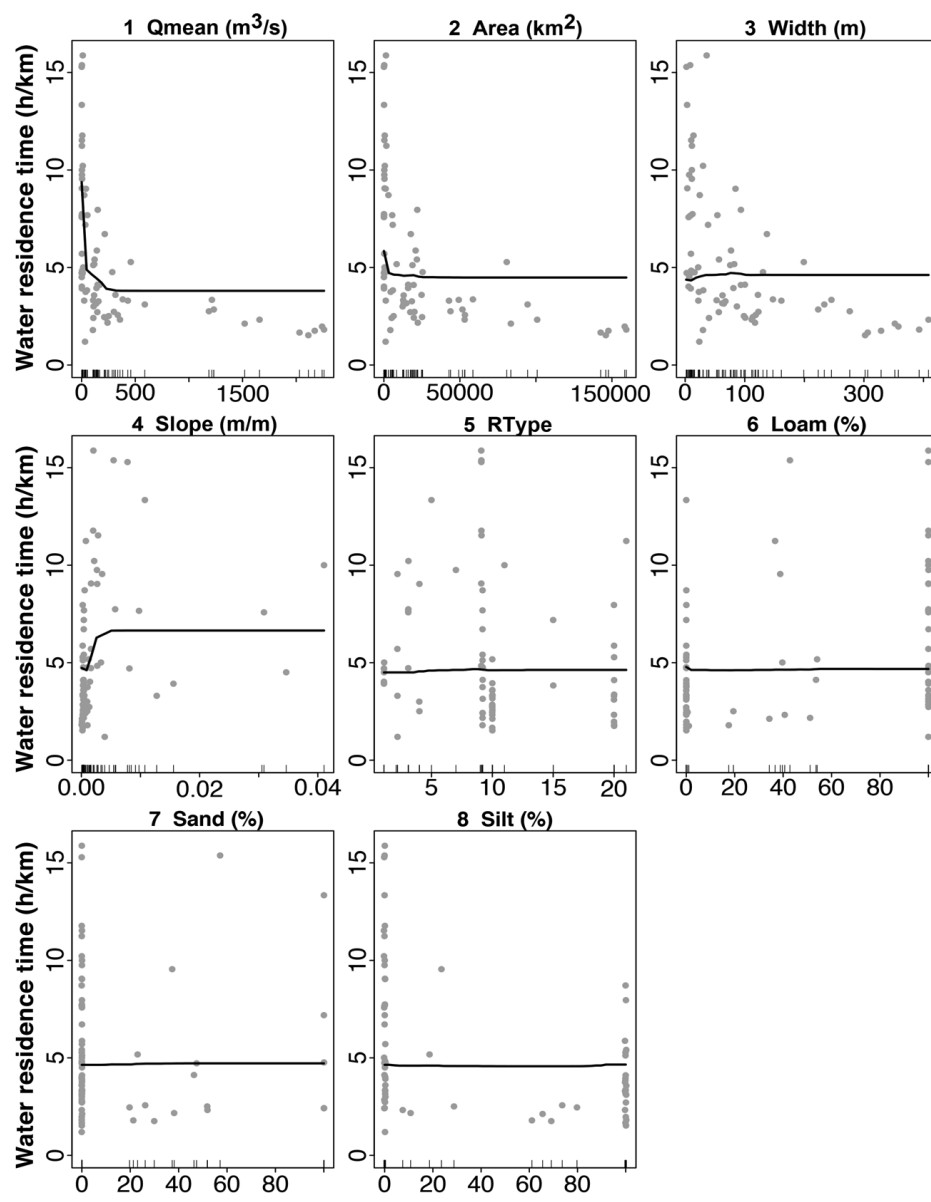

**Figure 3: Partial dependence plots showing the dependence of water residence time on hydro-geomorphologic variables after accounting for the average effects of all the other predictors in boosted regression tree analysis. Each point represents an observed value with rug plots at the bottom of each panel. Y-axes are predicted values of the fitted functions (WRTpred). All panels are plotted on the same scale for comparison.**





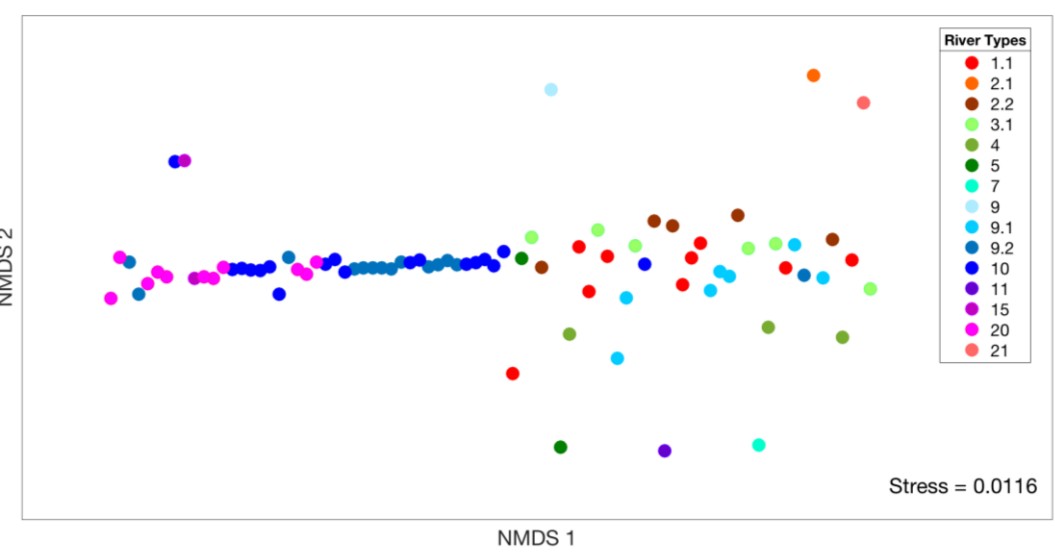

**Figure 4: Dissimilarities of the calculated WRT (h/km) in the Nonmetric Multi-Dimensional Scaling (NMDS) ordination space.**

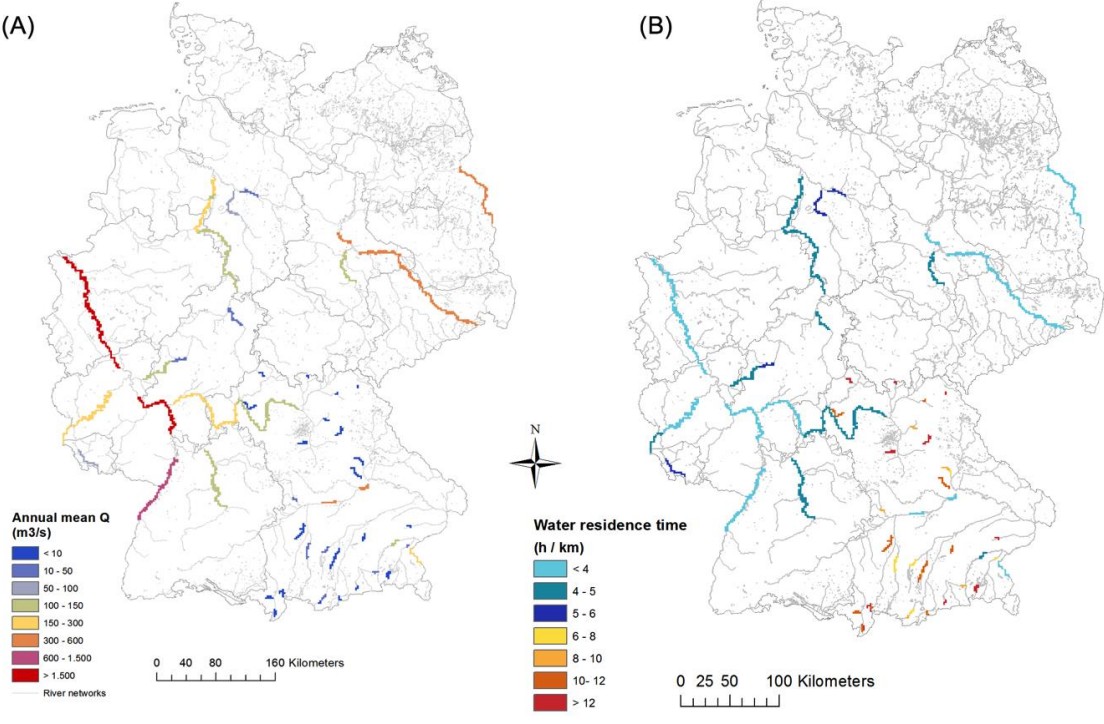

**Figure 5: Spatial distribution of annual mean discharge conditions during 2008-2013 (A), and predicted water residence time (h/km) for studied river reaches.**




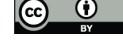


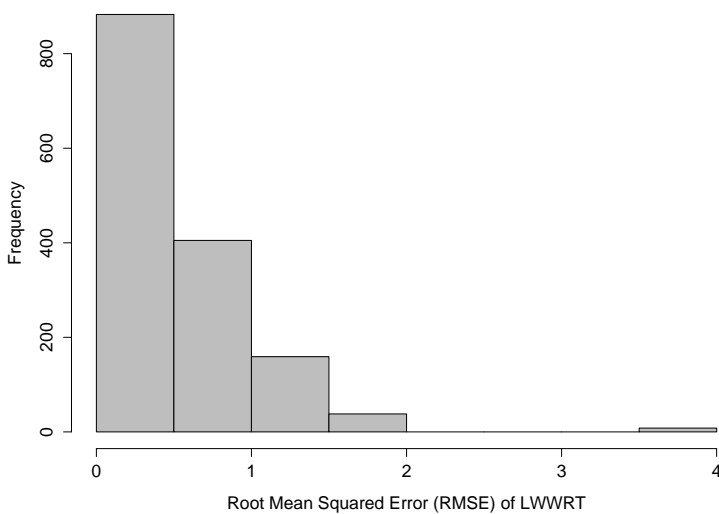

**Figure 6: Frequency distribution of RMSE of predicted length weighted water residence time (h/km) against observed values across all sites.**

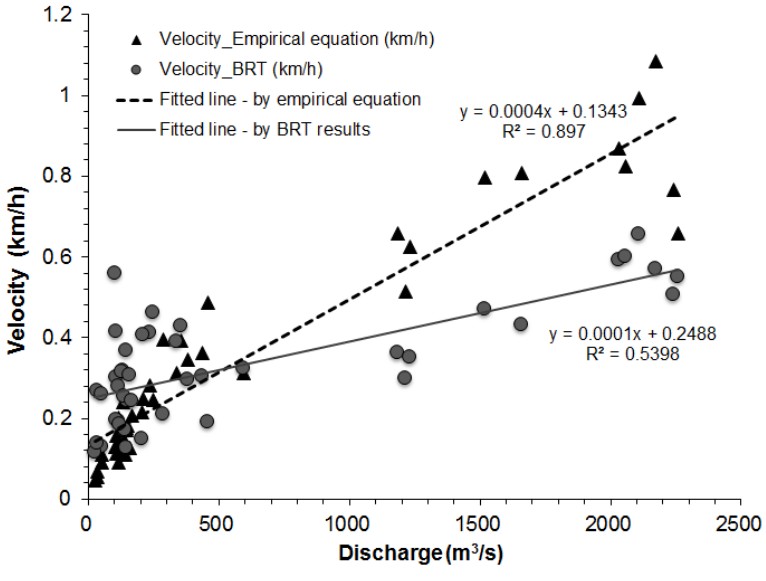


**Figure 7: Comparison of the fitted linear relationship between the average discharge (x-axis) and predicted flow velocity (km/h) by equation of Graf (1986), and the BRT model, respectively.**





**Figure 8: Statistical comparison of the mean discharge (in cubic meters per second) and corresponding water residence time (in hour per kilometer) in June 2013 (left), November 2011 (middle), and the difference between them (right), respectively.**




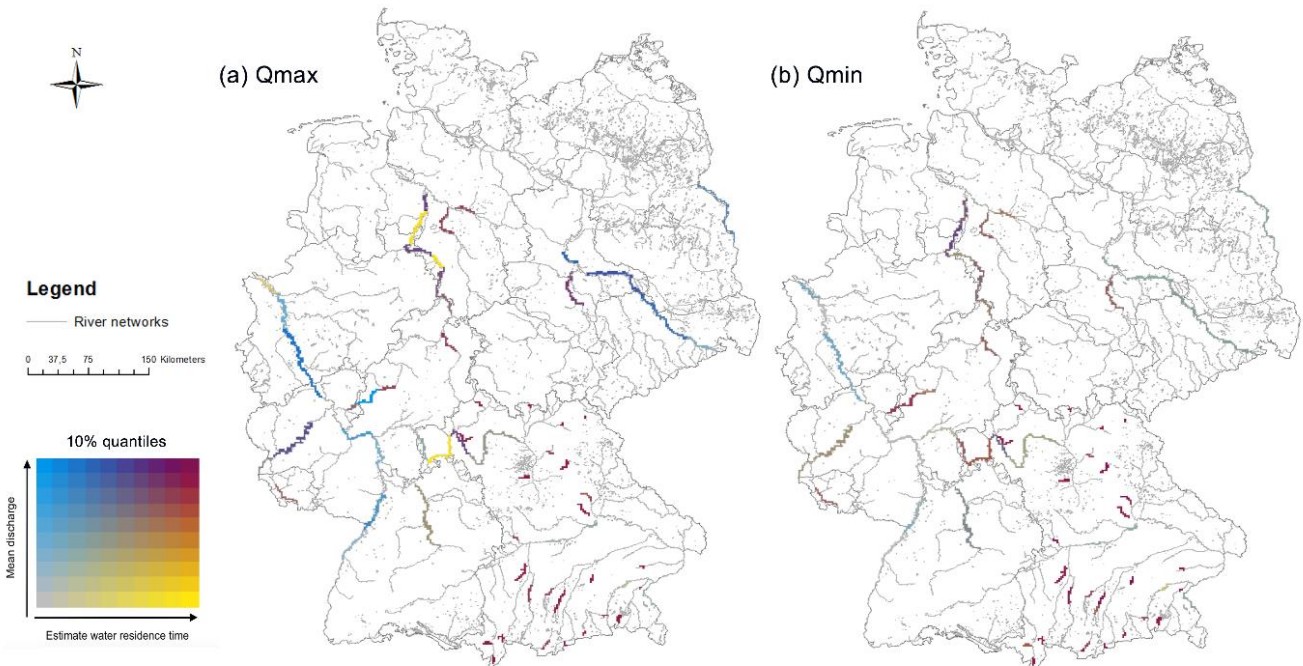

**Figure 9: Bivariate map of predicted water residence time (h/km) for (a) floods in June 2013 with Qmax scenarios and (b) drought with Qmin conditions during November 2011.**






**Figure 10: Two-dimensional interaction effects between the mean discharge (x-axis) and drainage area (y-axis). Colored scales are the estimated water residence time (h/km) accordingly.**




**Figure 11: Two-dimensional interaction effects between the mean discharge (x-axis) and river types (y-axis). Colored scales are the**
**estimated water residence time (h/km) accordingly.**






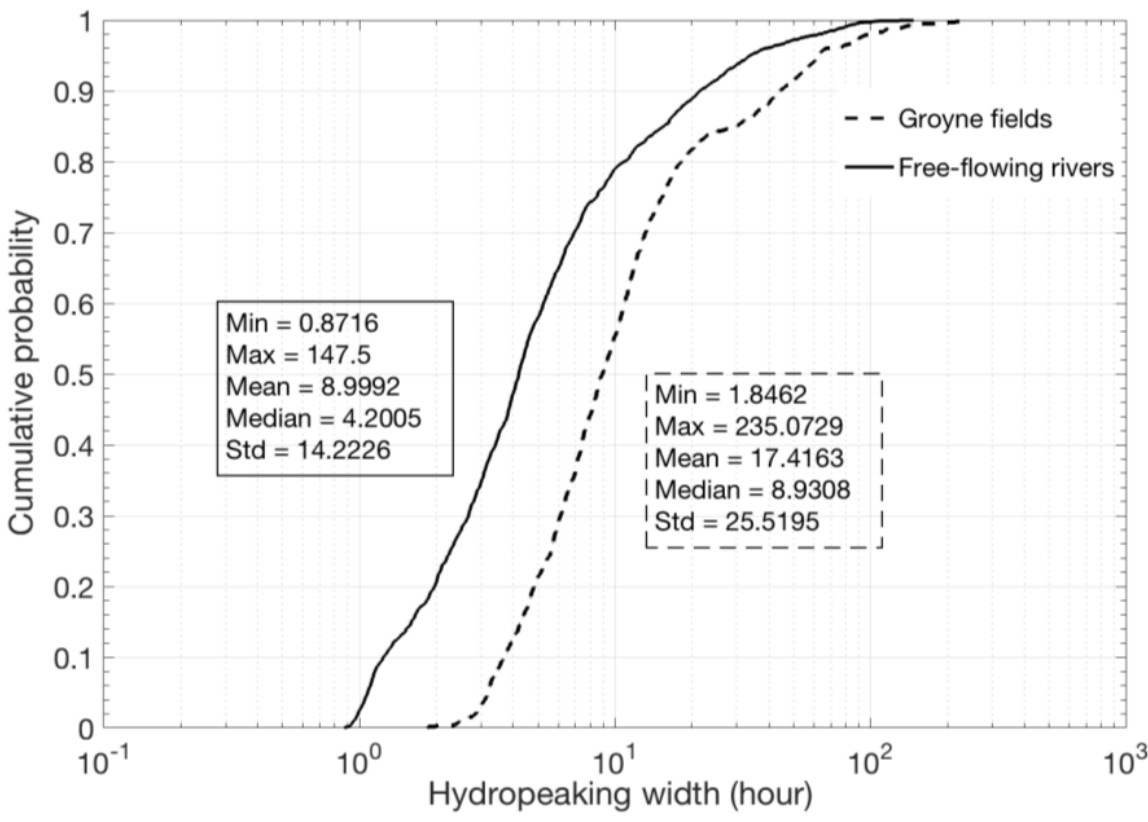


**Figure 12: Cumulative probability plots of the average hydraulic waves half-prominence widths (in hours) at river reaches with groyne fields and the free-flowing ones.**





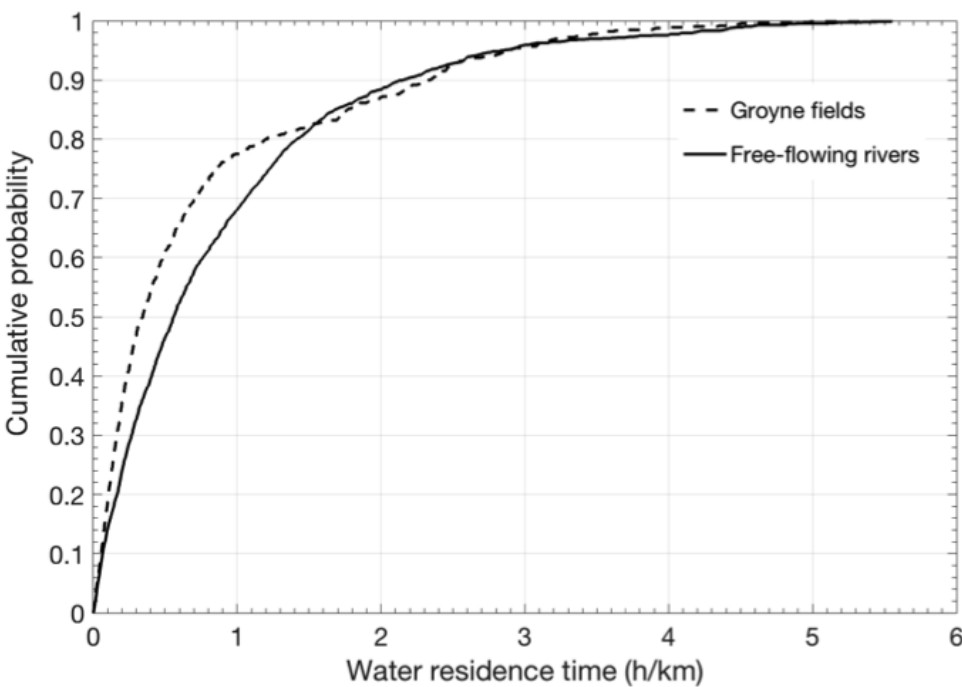

**Figure 13: Cumulative probability plots of the estimated water residence time (h/km) for river reaches with groyne fields and the free-flowing ones.**






**Table 1: Hydrologic and geographic variables of studied river reaches.**

| Continuous variables | Mean | Range | Std dev |
|---|---|---|---|
| Length (km) | 30.8 | 1.01 - 145.4 | 30.34 |
| Slope (m/m) | 0.00379 | 0.00005, 0.04104 | 0.00776 |
| Width (m) | 88.40 | 1.73 - 408.42 | 105.91 |
| Drainage area (km$^2$) | 25115.28 | 11.15 - 159427.5 | 41625.53 |
| Mean discharge (m$^3$/s) | 327.86 | 0.253 - 2259.32 | 610.45 |


**Table 2: Stream types covered by our study reaches (Pottgiesser and Sommerhäuser, 2004).**

| Main category | Sub-category |
|---|---|
| Alps and Alpine foothills | 1.1 = Small and mid-sized rivers |
| | 2.1 = Small rivers in the alpine foothills |
| | 2.2 = Mid-sized rivers in the alpine foothills |
| | 3.1 = Small rivers in the Pleistocene sediments of the alpine foothills |
| | 4 = Large rivers in the alpine foothills |
| Central highlands | 5 = Small coarse substrate dominated siliceous |
| | 7 = Small coarse substrate dominated calareous highland rivers |
| | 9 = Mid-sized fine to coarse substrate dominated siliceous highland rivers |
| | 9.1 = Mid-sized fine to coarse substrate dominated calcareous highland rivers |
| | 9.2 = Large highland rivers |
| | 10 = Very large gravel-dominated rivers |
| Central plains | 15 = Mid-sized and large sand and loam-dominated lowland rivers |
| | 20 = Very large sand-dominated rivers |



|  |  |
|---|---|
| Ecoregion independent streams | 11 = Small organic substrate-dominated rivers |
|  | 21 = Lake outflows |

Catchment size class:

|  |  |
|---|---|
| Small river: | 10 - 100 km$^2$ |
| Mid-sized river: | 100 - 1,000 km$^2$ |
| Large river: | 1000 - 10,000 km$^2$ |
| Very large river: | > 10,000 km$^2$ |

**Table 3: The relative influence of predictive variables of river hydro-geomorphology as computed from the fitted BRT model on water retention time.**

| Variable | Short name | Relative importance (%) |
|---|---|---|
| Mean discharge (m$^3$/s) | Qmean | 57.42 |
| Slope (m/m) | Slope | 21.54 |
| Drainage area (km$^2$) | Area | 15.64 |
| Mean river width (m) | Width | 2.41 |
| River type | RType | 1.25 |
| Substrate_Sand (%) | Sand | 0.70 |
| Substrate_Loam (%) | Loam | 0.69 |
| Substrate_Silt (%) | Silt | 0.34 |
| Substrate_Clay (%) | Clay | 0 |
| Substrate_Peat bog (%) | Peat bog | 0 |
| Substrate_Fen (%) | Fen | 0 |
