# Peer review of "Estimating water residence time distribution in river networks by boosted regression trees (BRT) model"

_Hydrology and Earth System Sciences, 2018_

## Referee Comment (RC1) · Anonymous Referee #1 · 4 Dec 2018

General Comments: This paper provides a new method, boosted regression trees (BRT) model, to estimate water residence time. 82 river reaches in Germany were used to assess the method by using an average of the 2008-2014 discharge data. Parallel studies for an extreme flood event and a dry month were assessed, although one main comment is that the specific analysis for these hydrological conditions needs to be clarified. Although this method has the potential to estimate water residence time more easily and with limited data, the advantages over current methods should be further emphasized within the text. Specific comments to clarify the methods and results and suggestions on how to redirect the discussion are included below in order to improve the overall message and strengthen the paper.

[Figure]

Specific Comments: 1. Clearly explain how this method is an advantage from current models to estimate water residence time and add this information throughout the text and in the abstract. a. At the end of introduction (L74) add the main goals of the paper and model analysis. L49-51 states that there is still a huge gap between detailed process-based models and over simplified empirical methods. State something similar at the end of the introduction and/or mention a main objective stated in the discussion (L190-191) that the author's aim to use predictive variables to facilitate the empirical estimation of WRT in a river on the basis of generally available information. b. Include in the abstract how the model was validated.

2. Improved explanations are needed of the main model equations, model inputs and outputs. Some specific examples include: a. Add a statement to justify the approach of model validation with the average discharge (2008-2014). b. L113-114 What information does this analysis provide? Although it may not be possible to fully describe the model, it is important to provide the reasoning for why and how partial dependence plots and fitted link functions for each variable operate and what information they provide. Basically, it is important to explain model inputs, a general description of what the model does and how it interprets and processes the data, and then a full description of the model output. The description on L115 is quite brief and it would be helpful to add here more information on how WRT is calculated. c. Equation 1 is WRT predict? This is stated on L152 but should be specified near to the equation to avoid confusion. d. L125 How is WRTobs originally calculated? This is important because it has some inherent bias in what the most important parameters/dependent parameters. e. L133, L138 (& previously in methods) further define Euclidean distance and "sum of all trees multiplied by the learning rate". Include a concise statement on each term and how each support the analysis. f. L182 states that parallel studies for extreme flood and dry months of 2 specific months. Was the same analysis conducted as for the average? Clarify that Figure 9 is WRTpred and if this value is compared to WRTobs as was done for the average.

3. Redirect the discussion to improve focus on the main results presented within the article: a. In order to link to the presented results, the following topics need to be further discussed 1) impacts of drought and flood events on WRT and implications of these hydrological conditions (i.e., extreme) and 2) how geomorphology attributes are more influential on small rivers and where the model capability was both best and worst at WRT estimates and why. The discussion should link to the main purpose of this analysis that is clearly stated at the end of the introduction (see comment #1). Elaborate on L198-199 WRT can be estimated even at low flows or based on this study only at high flows are accurate? b. Section 4.2 – it isn't obvious after reviewing the introduction through results why this is a main discussion point. Related to this point, the figures within the discussion should be removed or improved justification is required to include them but with a recommendation to move these into the results section.

---

## Referee Comment (RC2) · Anonymous Referee #2 · 18 Jan 2019

**Manuscript Review**

HESS 2018-309
Estimating water residence time distribution in river networks by boosted regression trees (BRT) model
Feng et al.

**Summary**

In their manuscript, Feng et al. explore the possibilities for estimating in-stream, reach-scale residence times in river networks by using a machine-learning approach (Boosted Regression Trees) to the analysis of spatial landscape data.  Water residence times were estimated at both a mean annual scale and during flood and under flood and drought conditions.   The authors found that river discharge, slope, and drainage area are the three primary contributing factors determining residence time.

The approach would in theory be valuable, as most studies of water residence times are based on process-based, deterministic models with high data demands, or on the results of field studies utilizing dissolved solute tracers.  The availability of a top-down methodology for determining residence times allows for more parsimonious approaches to watershed models and increases the potential for scaling up biochemical models to regional and continental scales.

However, in the current manuscript, it is not clear whether the approach has validity.  In particular, the methods section needs much clarification.  First, the authors compare "observed" with BRT-based predictions of water residence times, but it is unclear how those "observed" values are calculated or how valid they are.  An empirical equation is given, but with no information on the coefficients selected or the relationship between the empirical predictions and any validated, tracer-based results. Next, the authors have carried out their clustering analysis using an R package (dismo).  Their description of what is actually done with the package, however, is limited and not sufficient for the reader to understand the process or to duplicate the analysis.  A much better description of methods is needed.

Finally, I find the "averaging" approach here used for the reach parameters somewhat troubling.  The reaches, as defined here, are simply stretches of river lying between arbitrarily located monitoring points.  The distances range from 1-145 km.  The authors use mean values for all of their parameters, for all of the reaches, but of course while these mean values might represent actual conditions for a short reach, they could represent a variety of heterogeneous conditions across a larger area.  The authors should address this issue of averaging across areas very different size and make a case for why this approach is valid.

**Specific Comments:**

What is the validation approach?

Lines 101-102   Can you clarify your use of the term "reach" here.  You are trying to classify these reaches using the BRT approach, but because you are using "reaches" that are defined simply by an arbitrary length of river with gauging data at the upper and lower boundaries means that there could be very heterogeneous conditions along the length that you are considering. A simple additional discussion of how you define the reach and how you deal with the issue of heterogeneity along the stream corridor would be helpful here.  In addition, it would be useful to include some quantitative discussion of reach length (e.g. mean and median lengths).

Lines 104-105   What are these 13 types?  What is this based on?  Please clarify.

Lines 115-117   There would likely be strong correlations among these variables (mean discharge, drainage area, mean river width, length).  How do these correlations impact your analysis?

Lines 117-119   How relevant are these mean geomorphologic parameters if the reach is very long and the mean doesn't actually represent any portion of the river corridor very accurately?  This question needs to be addressed to make the subsequent analysis convincing.

Table 1   Decimal point values are unnecessary for most of these values—too many significant figures.

Lines 148-149   Your meaning here is unclear. What are you referring to when you mention "the known average values of predictive datasets for the complete river networks"?

Lines 144-158   Your methods here should be clearer.  It is very difficult to understand what was actually done in the analysis.

Line 163   What is "WRTpred"?  Define your variable here.

Lines 162-171   It is very unclear here how you calculated the "observed" water residence times.  You provide the empirical equation, proposed by Graf, but what coefficients are you using?  This needs clarification.

Line 179   When you refer to the Euclidean distance here, you should make clear that you are referring to the distance in the ordination space.

Lines 181-183   You are concluding here that the river reaches cluster in agreement with size. This seems to be true, but then you are using river size, as well as multiple attributes that would scale with river size (mean discharge, drainage area,

mean river width, length) to create the clusters.  This seems like a very circular argument.  Can you clarify what the relevance of the clustering finding might be based on this approach?

**Minor Details:**

There are frequent problems with grammar, missing articles, and awkward sentence construction throughout the paper that should be fixed.  Some examples include:

Lines 67-69      "To improve the understanding of WRT as carrier and as driving force for instream processes, while considereing impacts hydro-morphological of of river channels characteristics…."

Lines 81-82      "…other nonlinear statistical approach such as the Boosted Regression Trees (BRT) is becoming to play a part in hydrodynamic studies…."

Lines 119-120    "Substrate class of the sediment type for each river reach is represented in percentage (100% all classes in sum) according to their length that falls into each class."

Lines 173-175    "The elaboration of the results is structured in a) the spatial dissimilarity of geomorphology and hydrological factors for the studied river reaches, followed by b) the results of relative importance of variables calculated by the BRT model."